# CPT1A Supports Castration-Resistant Prostate Cancer in Androgen-Deprived Conditions

**DOI:** 10.3390/cells8101115

**Published:** 2019-09-20

**Authors:** Molishree Joshi, Gergana E. Stoykova, Maren Salzmann-Sullivan, Monika Dzieciatkowska, Lauren N. Liebman, Gagan Deep, Isabel R. Schlaepfer

**Affiliations:** 1Department of Pharmacology, University of Colorado Anschutz Medical Center, Aurora, CO 80045, USA; Molishree.Joshi@cuanschutz.edu; 2Division of Medical Oncology, University of Colorado Anschutz Medical Center. Aurora, CO 80045, USA; gergana.stoykova@ucdenver.edu (G.E.S.); maren.salzmann-sullivan@cuanschutz.edu (M.S.-S.); lliebman@umich.edu (L.N.L.); 3Department of Biochemistry, University of Colorado Anschutz Medical Center, Aurora, CO 80045, USA; monika.dzieciatkowska@cuanschutz.edu; 4Department of Cancer Biology, Wake Forest Baptist Medical Center, Winston-Salem, NC 27157, USA; gdeep@wakehealth.edu

**Keywords:** *CPT1A*, prostate cancer, androgen receptor, histone acetylation, enzalutamide, dihydrotestosterone

## Abstract

Prostate cancer (PCa) is the most common cancer in men, and the global burden of the disease is rising. The majority of PCa deaths are due to metastasis that are highly resistant to current hormonal treatments; this state is called castration-resistant prostate cancer (CRPC). In this study, we focused on the role of the lipid catabolism enzyme *CPT1A* in supporting CRPC growth in an androgen-dependent manner. We found that androgen withdrawal promoted the growth of *CPT1A* over-expressing (OE) tumors while it decreased the growth of *CPT1A* under-expressing (KD) tumors, increasing their sensitivity to enzalutamide. Mechanistically, we found that *CPT1A-*OE cells burned more lipid and showed increased histone acetylation changes that were partially reversed with a p300 specific inhibitor. Conversely, *CPT1A-*KD cells showed less histone acetylation when grown in androgen-deprived conditions. Our results suggest that *CPT1A* supports CRPC by supplying acetyl groups for histone acetylation, promoting growth and antiandrogen resistance.

## 1. Introduction

Prostate cancer (PCa) is the most common cancer in men [1], and the global burden of the disease is rising. In the United States, PCa causes nearly 30,000 deaths and 230,000 new cases yearly, making the search for new therapeutic approaches imperative. The majority of PCa deaths are due to metastasis that are highly resistant to current hormonal treatments [2,3].

Androgen-deprivation treatment (ADT) is currently the standard treatment for PCa that is localized or metastatic. Despite initial benefits of the androgen removal, some patient’s progress to an advanced disease state called castration-resistant prostate cancer (CRPC). This is the lethal form of the disease with limited treatment options, and it is still dependent on the androgen and androgen receptor (AR) axis. In fact, a salient characteristic of CRPC is reactivation of AR signaling. Extensive research has shown that most AR-regulated genes (androgen-response hallmark genes) are re-expressed in CRPC, and several mechanisms by which AR activity is preserved have been reported [4,5,6]. Recently, improved AR antagonists, also called anti-androgens, have shown to extend survival [7,8], but full remissions are rare. The main problem with using these potent AR inhibitors is the emergence of resistance, driven by mechanisms such as AR amplification, mutations, and alternative spliced variants, among others [9]. Another aspect of ADT is the profound changes to body weight and the lipid profile, with increases in triglycerides, cholesterol, and lipoproteins [10,11]. The ability of CRPC to use this increased circulating lipid for growth and anti-androgen resistance is still unknown.

The role of lipids in PCa has been studied for the last few decades [12,13,14], and a connection between lipid metabolism and CRPC is starting to emerge [15,16]. However, the way CRPC uses lipid and its dependency on beta-oxidation is still unexplored. A key player in lipid oxidation is the rate-limiting enzyme carnitine palmitoyltransferase 1 (CPT1), which works by transferring long-chain fatty acids into the mitochondria for oxidation [17]. In PCa, inhibition of *CPT1A* (the liver isoform of CPT1) reduces viability of LNCaP and VCaP cells and tumors [18], and increases PCa sensitivity to the anti-androgen enzalutamide [19,20]. Other studies have also shown that lipid oxidation is important for cancer survival [21], resistance to aniokis [22], and activation of oncogenic pathways [23]. However, it is unknown how lipid burning in the mitochondria is associated with AR in the nucleus and how this relationship can promote CRPC.

Besides being a limiting step for fat oxidation and ATP generation, *CPT1A* is also necessary for generating metabolic intermediates to synthesize macromolecules like lipids and nucleic acids, which are both essential for cancer growth. A recent study in endothelial cell sprouting has shown that *CPT1A* is needed to generate acetyl-coenzyme A (acetyl-CoA), which enters the TCA cycle and generates metabolic intermediates needed for de novo nucleotide synthesis, leading to DNA replication and proliferation [24]. Another study using immortalized hepatocytes has shown that fat oxidation is a major carbon source for histone acetylation, modulating lipid metabolism and reprogramming gene expression [25]. More recently, *CPT1A* has been implicated in lymphangiogenesis, promoting the generation of acetyl-CoA and acetylation capacity of the histone acetyl transferase p300 to acetylate histones at angiogenic gene promoters [26]. Thus, recent studies point to *CPT1A* as a lipid metabolic enzyme with the potential to modify macromolecule synthesis as well as gene expression.

Since the relevance of increased *CPT1A* expression in advanced PCa is emerging [19], in this study, we investigated the role of *CPT1A* in castration-resistant cell and tumor models. We used genetic models of *CPT1A* knockdown (KD) and overexpression (OE) and challenged them with androgens or enzalutamide. Particularly, we focused on the association between *CPT1A* in the mitochondria and the androgen response in the nucleus, with the hope of elucidating the nature of their relationship and the possibility of exploiting it for better CRPC therapies.

## 2. Materials and Methods

### 2.1. Cell Lines and Reagents

LNCaP-C4-2 cells were purchased from the University of Texas MD Anderson Cancer Center (Houston, TX, USA). Cell lines (22Rv1) were obtained from the University of Colorado Cancer Center (UCCC) Tissue Culture Shared Resource (2014) (Aurora, CO, USA) and were authenticated by Single Tandem Repeat analysis. Cells were passaged in RPMI media containing 10% serum supplemented with amino acids and Insulin (Thermo Fisher, Waltham, MA, USA). MDV3100 or enzalutamide (Selleckchem, Houston, TX, USA) was dissolved as a 20 mM stock in DMSO. Stocks were kept at −20 °C until ready to use. Fatty acids were purchased from Sigma (St. Louis, MO, USA), resuspended in ethanol for a stock solution of 10 mM, and stored at −80 °C. For cell use, fatty acids were first conjugated to 10% albumin, and then applied to cell media at the indicated concentrations.

### 2.2. Clonogenic Growth, Migration, and Growth in Suspension Assays

Clonogenic growth assays were done by plating cells in 12-well plates (Light Labs Colorado, Aurora, CO, USA) in media supplemented with fetal bovine serum (FBS) or charcoal-stripped serum (CSS) and the indicated drug doses. Crystal violet stains were analyzed with ImageJ software (https://imagej.nih.gov/ij/), as previously described [19]. Migration was done using the Incucyte^®^ Live Cell Analysis System (Sartorius, Ann Arbor, MI, USA), and analysis performed according to manufacturer’s instructions. Cell growth in suspension was done by growing the cells in ultra-low cell attachment plates (Sigma Aldrich, St. Louis, MO, USA) for 2 weeks, and imaging the spheres with an Olympus IX83 scope (Tokyo, Japan) and DP74 camera (Tokyo, Japan), followed by ImageJ analysis.

### 2.3. Seahorse Metabolic Flux Analysis

Mitochondrial respiration was done at the molecular and cellular analytical core at the University of Colorado, using a Seahorse XFe96 Analyzer with 96-well plates (Santa Clara, CA, USA). We measured oxygen consumption rate (OCR) using the XF Cell Mito Stress test, which provides a standard method to assess mitochondrial function in live cells. This test uses Oligomycin, FCCP ((4-(trifluoromethoxy) phenyl) carbonohydrazonoyl dicyanide) and a mix of rotenone and antimycin A to modulate the electron transport chain. We plated 40,000 cells/well and treated them overnight with no serum, and supplemented the next day with 50 uM fatty acids and albumin two hours before the assay. Assay media contained no glucose nor bicarbonate, and the pH was adjusted to 7.4. As a control, we also supplemented the cells with albumin by itself, free of fatty acids. Analysis was performed with Seahorse Wave software (Agilent, Santa Clara, CA, USA).

### 2.4. Immunofluorescence and Microscopy

LNCaP-C4-2 cells were grown in 22 mm glass coverslips to 70% confluency. For imaging, coverslips were fixed with 4% Paraformaldehyde and permeabilized with perm/wash buffer from BD Biosciences (San Jose, CA, USA). Hybridization with primary antibodies was done using perm/wash solution in humidified chambers. For mitochondria staining we used MTC02 (Abcam 3298) at 1:500 dilution overnight, followed by a secondary TRITC-conjugated antibody at 1:1000 for 1 h. BODIPY™ 493/503 (Thermo Fisher, Waltham, MA, USA) was used at 1:1000 dilution for 10 min to stain lipid droplets. Phalloidin-Alexa 488 (Thermo Fisher, Waltham, MA, USA) was used at 1:1000 dilution to demarcate the cell perimeter and identify individual cells. DAPI was used to stain the nucleus. Images were taken with an Olympus FV1000 laser scanning confocal via a 100× UPlanSApo oil objective in the Advanced Light Microscopy Core at University of Colorado Denver, Anschutz Medical Campus (Aurora, CO, USA). Analysis was done with free software FIJI (https://imagej.nih.gov/ij/), using the ROI feature to identify individual cells and measuring the mitochondria (red) and lipid (green) channels. A minimum of 20 measurements were done per condition.

### 2.5. Lentivirus Preparation and Transfection

Lentiviral particles for shRNA and ORF (complete open reading frame) specific to *CPT1A* were prepared at the Functional Genomics facility at the University of Colorado AMC. The following shRNAs from the Sigma shRNA library were used: TRCN0000036279 (*CPT1A-*KD) and control shRNA (NTshRNA) was SHC202. For *CPT1A* overexpression (*CPT1A-*OE), we used the ccsbBroad304-00359 clone from the CCSB-Broad lentiviral library.

### 2.6. Reverse-Transcriptase-PCR

For RT-PCR analysis, cDNA was synthesized (Applied Biosystems, Foster City, CA, USA) and quantified by real-time PCR using SYBR green (BioRad, Hercules, CA, USA) detection. Results were normalized to the housekeeping gene RPL13A mRNA and expressed as arbitrary units of 2^−ΔΔCT^ relative to the control group. A list of primers used is shown in Appendix A.

### 2.7. Mouse Xenograft Studies

Mice were purchased from Charles Rivers Labs and surgically castrated before cell inoculations under the skin. We used NTsh (control for KD), *CPT1A-*KD, EV (control for OE) and *CPT1A-*OE cell lines from C4-2 and 22Rv1 models to generate the xenografts. NSG mice were used for the C4-2, and nu/nu mice for 22Rv1 cells. About 2 × 10^6^ cells were inoculated per flank, and tumor growth was monitored daily with calipers. For MDV3100 (enzalutamide or Enza) studies, tumors were generated the same way, and when they reached ~500 mm^3^, we initiated oral treatment with Enza at 20 mg/Kg every other day for 3 weeks. Tumor growth was calculated using the formula volume = (π × length × width^2^) / 6.

### 2.8. Western Blot Analysis

Protein extracts of 20 µg were separated on a 7.5% SDS-PAGE gel and transferred to nitrocellulose membranes, as described [19]. The same protein lysates were probed on different blots. Antibodies to all the histone acetylation marks were purchased from CST (Boston, MA, USA), and are shown in Appendix A. Band signals were visualized with the LICOR system (Lincoln, NE, USA).

### 2.9. Histone Isolation and Mass Spectrometric Analysis

Histones were isolated from cells grown in FBS or CSS supplemented with 50 uM C^13^-acetate (Sigma Aldrich, St. Louis, MO, USA) for 48 h. Isolation was done with a histone isolation kit (Abcam 113476, Cambridge, MA, USA), and then run in a 20% SDS PAGE gel to separate the histones from other nonhistone proteins. Excised histone gel pieces were destained in ammonium bicarbonate/50% acetonitrile and dehydrated in 100% acetonitrile. Mass Spectrometric (MS) analysis was done at the CU School of Medicine’s Biological Mass Spectrometry Core. Digestion, peptide isolation, and MS analysis are described in detail in the supplementary methods section. Scaffold (Proteome Software v4.8, Proteome Software Inc., Portland, OR, USA) was used to validate MS/MS-based peptide and protein identifications. Peptide identifications were accepted if they could be established at greater than 95.0% probability, as specified by the Peptide Prophet algorithm. Protein identifications were accepted if they could be established at greater than 99.0% probability and contained at least two identified unique peptides.

### 2.10. Statistics

Student *t*-tests or ANOVA tests were used to compare between groups, followed by post hoc tests when appropriate. Corrections for multiple comparisons were done with Holm–Sidak method, alpha = 0.05. Analysis was carried out with GraphPad Prism software (v7, La Jolla, CA, USA). All data represent mean ± SD unless otherwise indicated.

### 2.11. Clinical Data Analysis

Prostate cancer datasets, comprising 3313 patients/3542 samples in 16 studies (PMID: 22722839, 26000489, 26855148, 23622249, 22610119, 28068672, 26928463, 20579941, 25024180, 29610475, 28927585, 26544944, 25201530, 28825054, 29622463, 29625055, 29625050, 29625048, 29617662, 29596782) were queried for *CPT1A* amplification on cBioportal (cbioportal.org). Three datasets (TCGA Cell 2015, TCGA PanCancer, and TCGA Provisional) were not included in the analysis as they had overlapping data with other selected studies. Cancer study and cancer-type detailed summaries and survival outcomes have been reported here.

## 3. Results

### 3.1. Characterization of CRPC C4-2 Cells with CPT1A Knock-Down (KD) or Overexpression (OE)

In order to elucidate the role of *CPT1A* in CRPC, we generated LNCaP-C4-2 cells with decreased and increased *CPT1A* expression using lentivirus transduction. The KD cells were generated with the same shRNA that was recently used in LNCaP cells in a previous publication [18]. We chose the C4-2 cell line because it is an established model of the castration-resistant phenotype observed in the clinic. This line was derived from the hormone-sensitive LNCaP cells and was grown in castrated nude mice for more than 12 weeks [27]. Another important aspect is that this model lacks the tumor suppressor PTEN, which is commonly lost in prostate cancers [28]. Figure 1A,B show the *CPT1A* mRNA and protein of the C4-2 generated cell lines, including their respective controls. As expected from our previous studies [18,19], the KD cells showed less clonogenic growth (Figure 1C,D), but surprisingly, the OE cells showed a similar trend. We then performed migration studies to study if *CPT1A* is needed for cell motility. Only the KD cells were possible to analyze since the OE cells did not attach well, and it was therefore difficult to assess migration into the scratch gap (Figure 1E). Since the OE cells grew better in suspension, we performed growth suspension assays on low-attachment plates. We observed a modest but significant increase in the diameter of the OE spheres compared with controls (Figure 1F). Thus, the decreased clonogenic growth in the OE cells in a two-dimensional (2D) model (Figure 1C) was not observed in a forced suspension culture.

Since *CPT1A* resides in the outer membrane of the mitochondria, we imaged mitochondria (red) and lipid droplet content (green) of the cells (Figure 2). Overall, we did not see significant changes in mitochondria in both cell lines, but lipid droplets were significantly increased in *CPT1A* OE cells. On the other hand, there was a significant decrease in lipid droplets in the KD cells, perhaps associated with a less aggressive phenotype. Next, we studied if the ability to accumulate lipid and its oxidation were correlated in *CPT1A* OE cells. The ability to burn long-chain fatty acids like oleic and palmitate, which need *CPT1A* to get into the mitochondria as acyl-carnitines, was assessed with Seahorse mito stress test assays (Figure 2E). The OE cells had an increased basal oxygen consumption rate (OCR) for these common long lipids (first three points in graph) and increased maximal respiration rate (after FCCP addition). There were no changes observed when fat-free albumin vehicle controls were used (Appendix A). Furthermore, when we supplemented cells with medium-chain fatty acids, which are less dependent on *CPT1A* to enter the mitochondria, we also observed increased OCR, indicating an overall increased ability to burn lipid in the OE cells (Appendix A). Our previous work demonstrated that *CPT1A* KD cells have decreased ability to burn lipids [18,29].

### 3.2. The CPT1A KD cells Show Increased Response to Androgens, While CPT1A OE Cells Show a More AR-Independent Phenotype

Previously, we observed that LNCaP cells with *CPT1A* KD showed increased expression of androgen-response hallmark genes [19]. To test if this was still the case in CRPC cells, we examined the expression of several androgen-regulated genes in the C4-2 KD and OE cells. As shown in Figure 3, KD cells showed increased expression of these AR-regulated genes, while the OE cells did not. In fact, when we supplemented the cells with 100 pM DHT (dihydrotestosterone), we observed a significant increase in AR full length (5-fold) and variant ARv7 (9-fold) in the KD cells, but not in the OE cell (Figure 3C). To ascertain if these results could be recapitulated in other genetically different CRPC cells, we generated 22Rv1 *CPT1A* KD and OE cells (Appendix A). We chose the 22Rv1 cells because they represent a good model of CRPC that has increased AR and ARV7 expression, but it is PTEN positive, so the AKT pathway is not always activated like in the C4-2 cell line [30,31]. Thus, the 22Rv1 offer a complementary model to the C4-2 cells used in the studies above. The 22Rv1 KD cells grew less in 2D clonogenic assays, like the C4-2 cells, but this effect was not seen in the 22RV1 OE cells, which showed no significant changes in clonogenic or growth in suspension compared with control cells. The migration capacity of the 22Rv1 KD cells was also reduced, like in the C4-2 KD cells, Appendix A. Regarding the lipid-burning capacity, 22Rv1 OE cells also showed increased OCR by seahorse assays with long-chain fatty acids, suggesting a functional *CPT1A* overexpressed enzyme. This lipid-induced OCR was suppressed by *CPT1A* inhibition with etomoxir at the start of the assay (Appendix A). However, regarding AR expression, we did not observe the dramatic increases in AR-FL and ARV7 in the 22Rv1 KD cells compared with the C4-2 cells, suggesting fundamental differences in the androgen response in the 22Rv1 cells (Appendix A). Alternatively, the abundance of AR isoforms in the 22RV1 cells could make them refractory to large changes in AR. Lastly, lipid droplet and mitochondria stains were also different from the C4-2 cells, since the 22Rv1 cells had very few lipid droplets but abundant mitochondria compared with the C4-2 cells. Interestingly, less mitochondria were observed with both the KD and the OE cells compared with their respective controls (Appendix A).

Next, we studied the effect of androgen stimulation on clonogenic growth at increasing doses in the C4-2 and 22Rv1 cells (Figure 4). The most salient results were that both the C4-2 and 22Rv1 KD cells showed increased growth compared with controls in the presence of DHT (Figure 4A,D, respectively). On the other hand, C4-2 OE cells showed a significant decrease in growth (Figure 4B), whereas no significant changes were observed in the 22Rv1 OE cells. To examine the effect of the anti-androgen enzalutamide in the OE cells, we studied their growth in the presence of increasing doses of enzalutamide. The C4-2 OE cells showed significant resistance to enzalutamide compared with controls (Figure 4C), while the 22Rv1 OE cells showed a more modest drug resistance trend (Figure 4F). A previous report by us [19] already showed that KD cells have increased sensitivity to enzalutamide in vitro.

### 3.3. CPT1A Expression in Tumors Modulates Growth and Response to Enzalutamide in Castrated Mice

The response of the KD and OE cells to enzalutamide in vitro inspired us to evaluate the growth of KD and OE cells in a castrated mouse model, mimicking the environment of the CRPC in the clinical setting (Figure 5). The C4-2 and 22Rv1 KD cells showed a modest decrease in tumor growth over time (Figure 5A,C, respectively). However, the C4-2 and 22Rv1 OE tumors showed significant increased growth (Figure 5E,G, respectively). These results underscore how the absence of androgens promotes the growth of the *CPT1A OE* tumors. Next, we challenged a new set of well-established tumors with oral enzalutamide for approximately 3 weeks. The C4-2 *CPT1A* KD cells showed a significant decrease in growth compared with control tumors (Figure 5B), and this effect was also observed in the 22Rv1 KD model, albeit more modestly (Figure 5D). Regarding the C4-2 and 22RV1 *CPT1A* OE cells, both tumor models showed increased growth compared with controls (Figure 5E,G, respectively). Treatment with enzalutamide did not decrease the growth of the OE cells, in fact it seemed to enhance the growth of the tumors (Figure 5F,H). Thus, in the absence of androgens and with AR blockade, *CPT1A* overexpression provides a growth advantage to the tumors.

### 3.4. CPT1A Expression Modulates Histone Acetylation in An Androgen-Dependent Manner.

Studies in liver cells have shown that fat oxidation can provide up to 90% of acetyl carbon for histone acetylation and gene expression modulation [25]. In order to identify possible epigenetic changes that could mediate the connection between *CPT1A* and AR, we examined the acetylation status of the C4-2 cells (Figure 6). We examined the lysine acetylation pattern of whole cell lysates of C4-2 *CPT1A* KD and OE cells in FBS (castrate levels of androgens, [32]) and CSS (androgen-deprived) conditions (Figure 6A). We observed increased global lysine acetylation in the CSS condition, especially in the OE cells, likely due to the increased *CPT1A* activity and acetyl-CoA production. Next, we studied histone acetylation patterns by western blot (Figure 6B). A Coomassie stain of the isolated histones is shown in Appendix A. The most dramatic difference was observed in Histone 3 Lysine 9 acetylation (H3K9ac); the OE cells showed increased acetylation in CSS compared with controls, but the opposite was seen under FBS conditions. In contrast, there was a significant increase in acetylation of H3K9 in KD cells in FBS, but a dramatic reduction of the mark in CSS conditions compared with the control lysates. Other histone acetyl marks like H3K14, K18, K27, and H4K8 also showed a similar trend of decreasing in KD and OE cells in FBS. Interestingly, we continued to see a decrease in acetylation of H3K14, K18, K27, and H4K8 in KD cells in CSS conditions, while the OE cells showed either no change or modest increase (exception in H4K8), reflecting global acetylation changes. These data point to a modulatory role of androgens on *CPT1A* activity and downstream acetylation events. Additional histone acetylation marks are shown in Appendix A.

To study the labeling of histones according to *CPT1A* expression by mass spectrometry, we treated cells grown in CSS with C^13^-acetate to follow the fate of the acetate carbons in histones. Acetate is carbon form that is readily picked up by tumor cells and significantly contributes to the pools of acetyl-CoA in the cell [33]. Figure 6C shows the overall acetylation for Histone 3 and Histone 2B in the C4-2 cells, as well as all the C^13^-labeled histones. The H3 and H2B were the histones in which we observed the most acetylated peptide counts. Less numbers of C^13^-labeled histone peptides were observed, but they seemed to follow the same pattern as the non-labeled ones. When the same experiment was conducted in the presence of A485, a potent selective inhibitor of p300 and CBP that competes with acetyl-CoA [34], we observed a global decrease in histone acetylation under the same conditions, suggesting that de novo C^13^-acetylation of histones was taking place in CSS conditions.

To study the relevance of *CPT1A* expression in advanced PCa, we mined the cBioportal database that now includes more metastatic and aggressive tumors. Figure 7A shows all PCa studies with *CPT1A* amplification data; a total of 3542 samples from 3313 patients across 16 studies were queried for *CPT1A* amplification. *CPT1A* was amplified in 4% (116) patients and 4% (124) samples. Classification of this same data by PCa types clearly showed that *CPT1A* was amplified in 21.43% of CRPC and 16.67% of neuroendocrine carcinomas compared with 2.99% PCa adenocarcinomas (Figure 7B), suggestive of a correlation between *CPT1A* overexpression and more aggressive prostate cancer. We further evaluated this data set for overall patient survival outcome (patient survival data was available for only 484 patients). *CPT1A* was amplified in 17 out of 484 patients, and of these 17 patients, 13 were deceased with a median survival of 17.6 months. This was statistically significantly lower (*p* = 8.52 × 10^−11^) compared with the cohort of patients with no alteration in *CPT1A*; this group consisted of a total of 467 patients, including the 120 deceased, whose median survival time was 120 months. Overall, the Kaplan Meier curve suggested that patients with amplified *CPT1A* had significantly poorer outcome. Due to the small sample size and insufficient data for each sample, we were not able to parse this further to look for links among *CPT1A* amplification, cancer type, and survival outcome.

## 4. Discussion

Understanding lipid metabolism in CRPC can open doors to new and improved therapies. In this report, we focused on the role of *CPT1A* in supporting CRPC growth in an androgen-dependent manner. We found an inverse relationship between *CPT1A* expression and androgen responsiveness in our models. Specifically, androgen withdrawal promoted the growth of *CPT1A*-OE tumors while it decreased the growth of *CPT1A*-KD tumors and increased their sensitivity to enzalutamide. This association is important, since CRPC remains lipid- and AR-dependent [35], but the use of potent anti-AR drugs like enzalutamide have a limited response and virtually all patients fail and develop resistance [7,8]. Our results open opportunities to target lipid oxidation in the mitochondria and AR signaling in the nucleus simultaneously, as a more effective therapy or perhaps a more durable response to anti-androgens.

Since *CPT1A* is not an oncogene, but a liver enzyme also abundant in epithelial cancers [36], finding its role in cancer has not been intuitive, especially since most cancers have a strong dependency on glucose and rely on the Warburg effect [37]. In the case of PCa, there is more dependency on lipids for growth and survival, which makes the FDG-PET imaging not effective in diagnosis and staging [29]. However, successful tumors that survive the metastatic process and drug treatments become more resilient and adapted to harsh environments. They do this by increasing their dependency on metabolic enzymes like *CPT1A*, which can lead to production of ATP, antioxidants, and metabolic intermediates that sustain growth and drug resistance [22,23,38,39]. Recently, as the PCa databases have become more complete, *CPT1A* is emerging as a potential biomarker of patient survival in advanced PCa. The present study provides clues as to how *CPT1A* contributes to CRPC and AR activity.

Generating *CPT1A*-OE cells was challenging since they came off the plates very easily, indicating a tendency to grow better in suspension. The accumulation of lipids observed in CPT1A OE cells is also a characteristic of more aggressive cancers [40,41,42]. Since these cells also had more lipid-burning capacity, it points to the possibility of a lipid cycle where CRPC cells accumulate lipid stores that burn quickly, like a seemingly futile cycle, but one that promotes survival. In fact, targeting both lipid synthesis and lipid oxidation simultaneously could be an interesting approach to decrease CRPC growth and progression [18].

The increased growth of the *CPT1A*-KD cells under androgen stimulation suggested the modulatory role of androgens in *CPT1A* activity and revealed a reciprocal association (Figure 4) that could be therapeutically exploited. In fact, in castrated mice, *CPT1A*-KD tumors grew less and were more sensitive to enzalutamide, which could be due to a compensatory upregulation of the androgen response to try to restore lipid beta-oxidation. This upregulation could make the cells more sensitive to androgens for growth, and more sensitive to the anti-androgens. On the other hand, *CPT1A*-OE cells and tumors showed the opposite effect; growth inhibition with androgens and growth stimulation with androgen depletion (Figure 4 and Figure 5). These results suggest that *CPT1A* expression is important for growth in androgen-depleted conditions, but remains modulated by androgens since increasing doses of DHT decreased growth of *CPT1A*-OE cells (Figure 4). We cannot rule out the possibility that *CPT1A*-OE cells remain sensitive to very small doses of androgens, with a very low threshold of AR activity. This phenomenon has been observed by reducing prohibitin in PCa cells, resulting in increased histone acetylation and sensitization cells to low levels of androgens, [43]. The possibility that these low levels of androgens are generated from cholesterol stored in lipid droplets is also very plausible [44], and remains to be studied in *CPT1A*-OE cells. These results may have implications for recent trials where androgen supplementation was used to treat CRPC by exploiting the androgen-repressive functions at supraphysiological levels [45]. Our results suggest that *CPT1A* could be a biomarker to stratify patients that will respond to such hormonal challenges.

In this work, we offered a potential pathway of communication between *CPT1A* activity and nuclear acetylation changes that could explain the growth phenotypes we observed. The increased lysine acetylation in OE cells in castrated (CSS) conditions was expected since excess acetyl-CoA is known to acetylate cytoplasmic proteins [46]. Of note was the enhanced acetylation of H3K9 in the OE cells in androgen-deprived conditions. This data is further supported by the fact that H3K9ac accompanied by presurgery PSA level has been reported to predict the risk of the biochemical recurrence after radical prostatectomy [47]. However, the dramatic decrease in histone acetylation in KD cells was unexpected (Figure 6). We speculate that this is due to a deficient supply of acetyl-CoA via *CPT1A* in the absence of androgens, decreasing growth. Another possibility for this decrease in histone acetylation in the KD cells is a compensatory activity of HDACs to try to activate AR signaling [48], or perhaps to free-up acetyl groups for energy purposes. Interestingly, in the presence of androgens (FBS), the KD cells had a dramatic increase in H3K9ac signal, suggesting a priming effect of the androgens to stimulate growth, which is what we observed with DHT supplementation (Figure 4). On the other hand, acetylation marks increased in *CPT1A*-OE cells in ADT conditions, a situation that has been recently associated with CRPC using high-throughput peptide arrays [49], where hyper-acetylation of Histone 3 marks and increased p300 activity in CRPC was observed. Our results suggest that *CPT1A* supports CRPC by supplying acetyl groups for histone acetylation, promoting growth and anti-androgen resistance. The specific genes modified by these acetylation changes await to be discovered. Figure 8 shows our proposed model for the role of *CPT1A* overexpression in *CRPC*.

Lastly, in an era of increased utilization of immunotherapies, the role of immune cell metabolism is rapidly increasing. In fact, T cells use glycolysis to expand and activate, while lipid oxidation via *CPT1A* keeps them in a suppressed state, unable to attack tumors [50]. These results strongly suggest that metabolic reprograming of the T cells in tumors is a novel Achilles’s heel that could be used to re-activate the immune system and invigorate antitumor immunity. The recently reported PROCEED trial including metastatic CRPC patients exposed to sipuleucel-T and enzalutamide [51] opens the door for studying novel sequencing of therapies that could potentially include targeting of *CPT1A* for optimal treatment.

## Figures and Tables

**Figure 1 cells-08-01115-f001:**
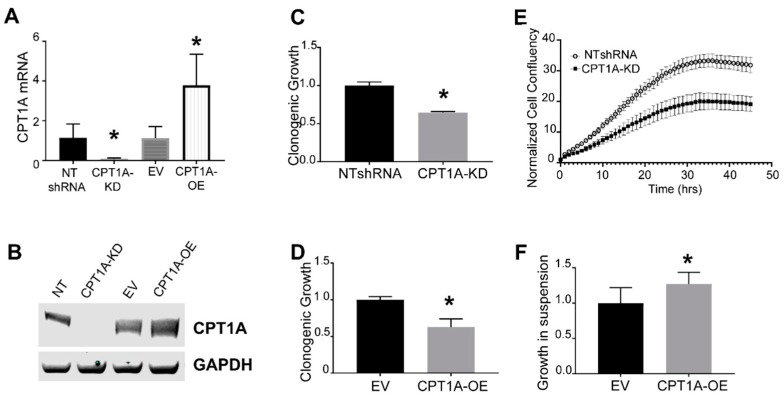
Characterization of C4-2 cells with *CPT1A* knock-down (*CPT1A*-KD) or *CPT1A* overexpression (*CPT1A-OE*). (**A**): *CPT1A* mRNA levels were measured in *CPT1A* knockdown and overexpressing C4-2 cells using qRT-PCR, (n = 4 per group. Students *t*-test; **p* = 0.002 for KD and **p* = 0.007 for OE). Each measurement was normalized to corresponding RPL13A mRNA. (**B**): Representative western blots showing the protein level of *CPT1A* in *CPT1A-*KD, *CPT1A-OE* and their respective controls (NTshRNA, EV) in C4-2 cells. GAPDH was used as loading control. (**C**,**D**): Clonogenic growth of *CPT1A*-KD (**C**) and *CPT1A*-OE (**D**) compared with their respective controls (n = 4 per group, Students *t*-test. * *p* < 0.01 vs. controls). (**E**): Scratch wound / migration assay on Essen IncuCyte^®^. Cell migration was measured by cell confluency (n = 10 per group). The scratch area at time 0 was set to 1. Representative images are in Appendix A. Similar experiments with *CPT1A*-OE cells did not yield usable data, as they did not adhere firmly, resulting in nonuniform scratch lines that could not be measured reliably. (**F**): Growth assay in suspension conditions with *CPT1A*-OE cells and control cells. *CPT1A*-OE and control EV cells were plated in low-adherence plates in complete media with serum for 14 days; n = 8 per group, * *p* = 0.014 vs. control. Representative images in Appendix A.

**Figure 2 cells-08-01115-f002:**
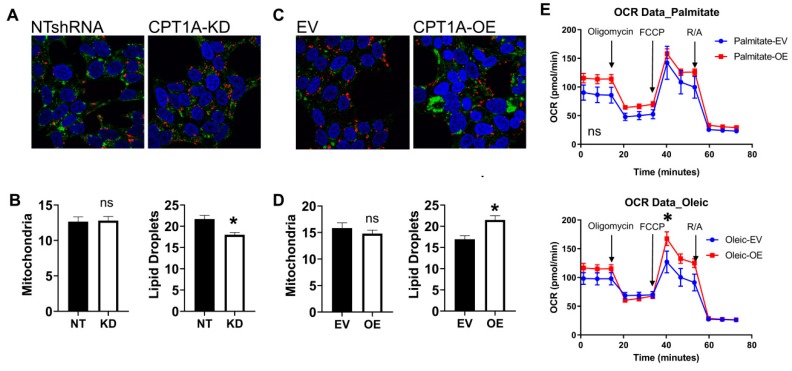
Phenotypic characterization of C4-2 *CPT1A*-KD and *CPT1A*-OE cells. (**A**–**D**): Mitochondrial and lipid droplet staining in *CPT1A*-KD cells (**A**,**B**) and *CPT1A*-OE cells (**C**,**D**). Cells were stained for lipid droplets (green), mitochondria (red), and nuclei (blue). A minimum of 20 cells per group were captured and quantitated using ImageJ. * *p* < 0.01 vs. control. (**E**): Oxygen Consumption Rate (OCR) of C4-2 *CPT1A*-OE and control EV cells (n = 10 each) was measured using the Seahorse XFe96 Analyzer. Cells were serum-starved overnight and then incubated with the BSA-conjugated fatty acids (palmitate or oleate) 1 h before the assay. Traces shown correspond to Cell Mito Stress protocol, using Oligomycin (2 uM), FCCP (2 um) and rotenone/antimycin A (1 uM) at 20, 40, and 60 min respectively, *p** = 0.012 compared with EV control. Seahorse graphs show mean ± SEM.

**Figure 3 cells-08-01115-f003:**
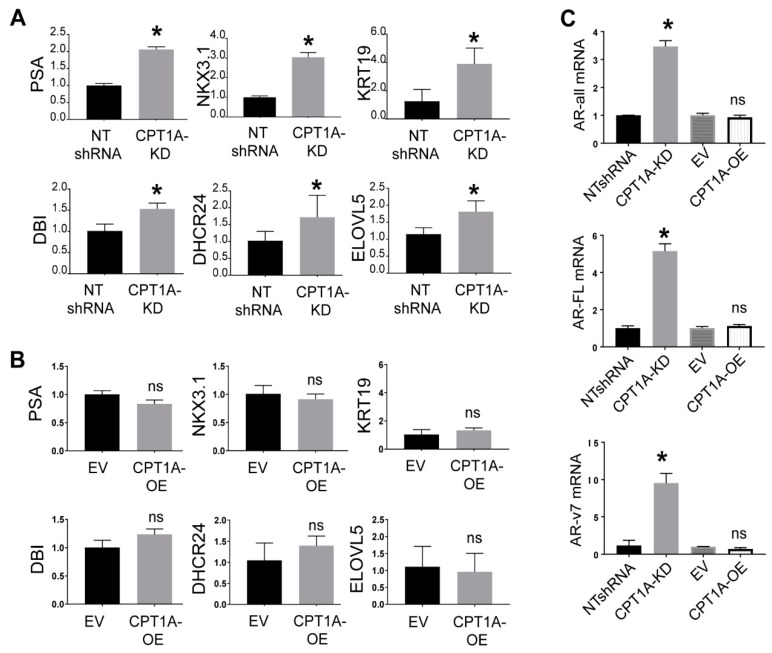
AR response in C4-2 *CPT1A*-KD and *CPT1A*-OE cells. (**A**): Basal expression of AR response genes in C4-2 *CPT1A*-KD and control NTshRNA cells, n = 4 per group; Students *t*-test. * *p* < 0.01 vs. control. (**B**): Basal expression of AR response genes in C4-2 *CPT1A*-OE and control EV cells. Data was normalized to RPL13A mRNA, n = 4 per group, changes were not significant. (**C**): AR mRNA expression in response to DHT. Expression of AR-all (measures all isoforms of AR), ARFL (full length AR) and ARv7 (AR variant v7) was measured in response to 100 pM DHT in *CPT1A*-OE and control EV cells, n = 4 per group. * *p* ≤ 0.01 compared with its respective control.

**Figure 4 cells-08-01115-f004:**
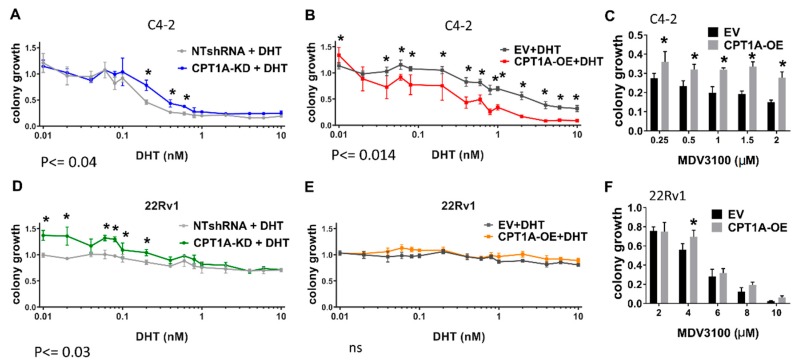
Clonogenic growth of *CPT1A*-KD and *CPT1A*-OE cells in the presence of androgens or anti-androgens. Cells were grown in media containing FBS supplemented with increasing DHT (androgen) or MDV3100 (enzalutamide) concentrations. Colonies were analyzed by normalizing to vehicle treatments. Significant results between conditions are indicated: (**A**–**C**): Clonogenic growth of C4-2 *CPT1A*-KD and control cells (**A**) and C4-2 CPT1-OE and control cells (**B**) in presence of increasing amounts of DHT, n = 4 per group. (**C**) Clonogenic growth of C4-2 CPT1-OE and control cells with increasing concentrations of MDV3100, n = 4 per group, * *p* < 0.01. vs. control. (**D**–**F**): Clonogenic growth of 22Rv1 *CPT1A*-KD and control (**D**) and 22Rv1 CPT1-OE and control (**E**) with increasing amounts of DHT, n = 4 per group. (**F**) Clonogenic growth for the 22Rv1 CPT1-OE and EV control cells with increasing concentrations of MDV3100, n = 4 per group, **p* = 0.01 vs. control.

**Figure 5 cells-08-01115-f005:**
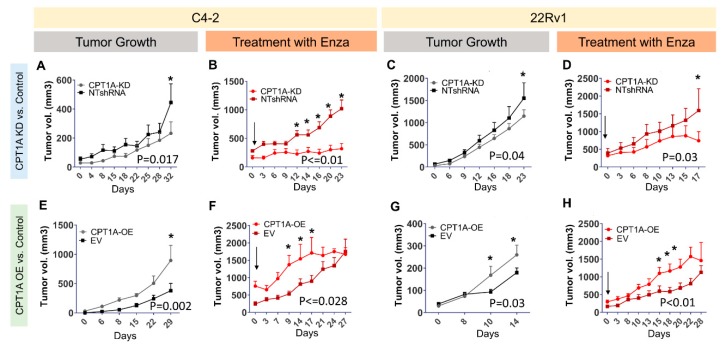
*CPT1A* expression in tumors modulates their growth and the response to enzalutamide in castrated mice. NSG mice were used for the C4-2 cells and nu/nu mice for 22Rv1 cells. Mice were castrated 2 weeks before cell implantations. Figures (**A**) and (**C**) show tumor growth curves for C4-2 and 22Rv1 with *CPT1A*-KD and control NT-shRNA cells, respectively. Similarly, Figures (**E**) and (**G**) show tumor growth curves for C4-2 and 22Rv1 with *CPT1A*-OE and EV control cells, respectively. In separate experiments, mice received oral gavage of 20 mg/kg enzalutamide (Enza) once tumors were established and growing (≥300 cc). Figures (**B**) and (**D**) show response to Enza in tumors with *CPT1A*-KD, whereas figures (**F**) and (**H**) show response to Enza in tumors overexpressing *CPT1A* (*OE*). Significance differences are indicated (*) for each group. Mouse numbers per graph: (**A**): (NT = 11, KD = 13), (**B**): (NT = 5, KD = 5), (**C**): (NT = 12, KD = 14), (**D**): (NT = 5, KD = 5), (**E**): (EV = 8, OE = 8), (**F**): (EV = 6, OE = 6), (**G**): (EV = 7, OE = 7), (**H**): (EV = 5, OE = 5).

**Figure 6 cells-08-01115-f006:**
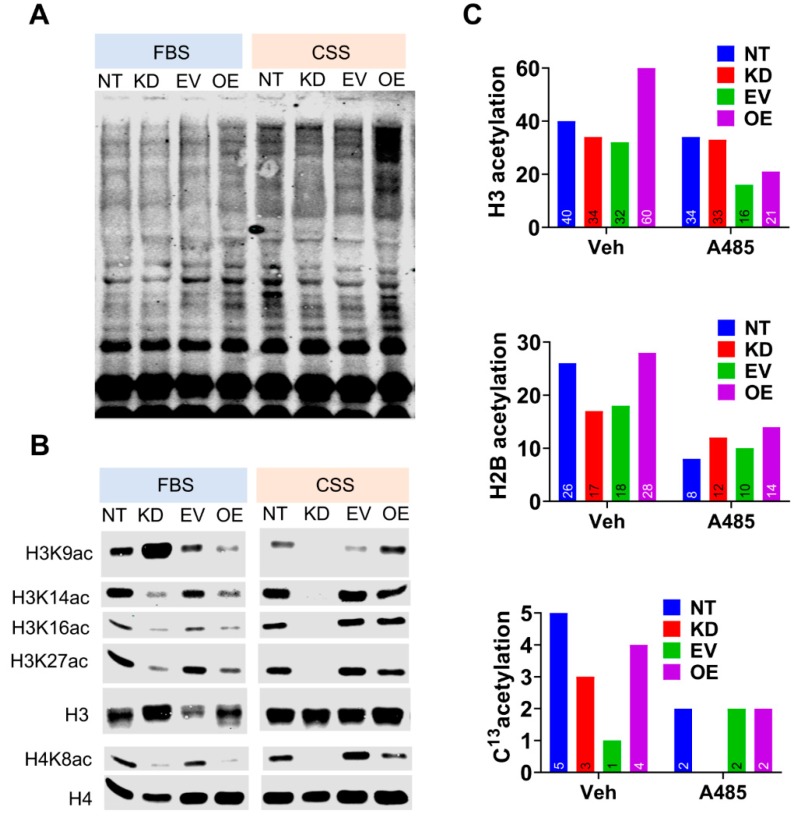
*CPT1A* expression modulates histone acetylation in an androgen-dependent manner. (**A**): Representative western blot of cell lysates with pan-acetylated lysine antibody. (**B**): Western blot analysis of acetylated histone lysine residues from C4-2 cells (KD and OE) cultured in FBS or CSS (steroid- free media). (**C**): Lysine acetylation of histones H3 and H2B, isolated from C4-2 cells cultured in CSS, with or without p300 inhibitor A485 (5 µM), and measured by mass spectrometric analysis. Additionally, cells were treated with C^13^-acetate for 24 h in CSS with or without A485, followed by histone isolation and mass spectrometric analysis. Numbers on bars indicate amount of specific 13C-acetylated histone peptides identified.

**Figure 7 cells-08-01115-f007:**
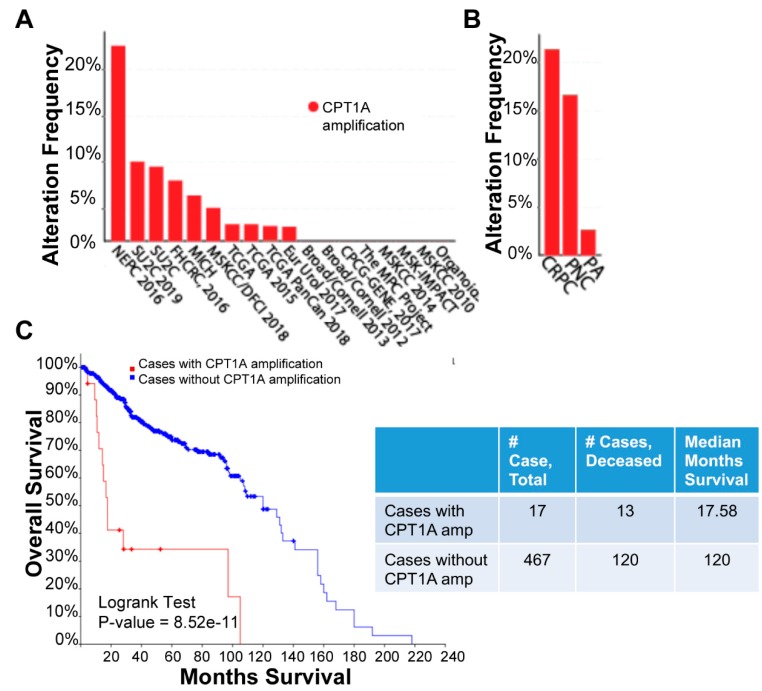
*CPT1A* amplification in prostate cancer across 16 publicly available studies. Data shown represents analysis of cBioportal database. A total of 3542 samples from 3313 patients across 16 studies were queried for *CPT1A* amplification (**A**): Graph shows frequency of *CPT1A* amplification across the studies. (**B**): Same data as in A, but reorganized according to PCa type: PA, prostate adenocarcinoma; NPC, neuroendocrine; CRPC, castration-resistant PCa. (**C**): Kaplan Meier curve showing overall survival for 484 patients. Red line shows cases with *CPT1A* amplification. Table shows the number of deceased cases and median months survival according to *CPT1A* amplification status.

**Figure 8 cells-08-01115-f008:**
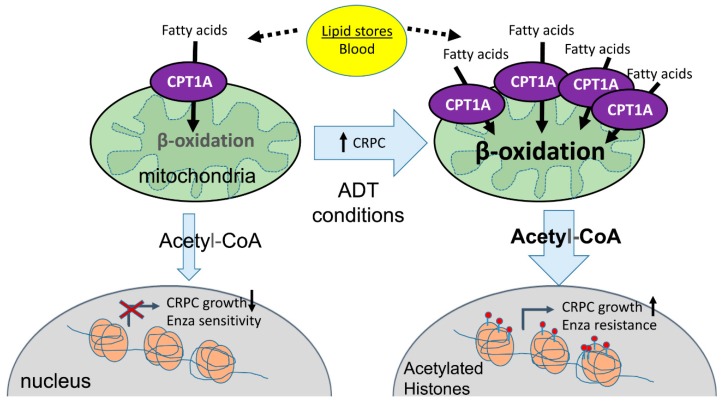
Proposed working model. In castrated conditions, increased *CPT1A* activity in the mitochondria increases beta-oxidation and generation of acetyl-coA. The excess of acetyl-CoA in the cytoplasm is used by the CRPC cells to acetylate histones and promote increased growth and resistance to anti-androgen therapies like enzalutamide, further promoting the castration-resistant phenotype.

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
