# Peer review of "CPT1A Supports Castration-Resistant Prostate Cancer in Androgen-Deprived Conditions"

_cells, 2019, doi:10.3390/cells8101115_

Round 1

Reviewer 1 Report

The Ms by Schlaepfer deals with the involvement of CPT1A enzyme, a key player in the lipid oxidation, in the prostate cancer resistance and progression. The link with the role of acetylated histone is fascinating, however, results are presented in a very confuse manner, lacking sometimes a scientific language (i.e. lines 203-206).

The most critical points lie in the Results section presentation: Results lack linearity and a sequence that facilitates reading. Sometimes data of the two cell lines are presented, sometimes only one, in a confused way (about cell lines, a justification of the reason why these lines were used and their descriptions must be added).

Minor point: the correct writing of LNCaP-C42 is C4-2.

The figures are too complex, sometimes the significance is missing.

Fig.1F: Are the Authors sure of   the statistically significant difference (p< 0.01) of bars?

Fig. 2E: Are the points statistically significant? It's not reported either in the figure or in the text

Fig. 6 must be divided into at least 3 different figures. The analysis on clinical data requires an independent figure. Concerning the survival curves of patients with CRPC depending on the CTP1A structure, it is necessary at least to separate neuroendocrine tumors from adenocarcinoma. Comments on this result (lines 258-265) are rather confused

The cartoon of the proposed study model must be reported in the Discussion and not in the Results.

In summary, Results need to be re-written, implementing data on cell lines

Author Response

Please see attachment, Thank you.

Reviewer 2 Report

In this manuscript, the authors investigate the role of CPT1A in the pathogenesis of prostate cancer. For this investigation, knockdown and overexpression experiments are done in two different cell lines. Phenotypical effects of knockdown and overexpression are then characterized in vitro as well as in a xenograft model. The authors also investigate sensitivity to the anti-androgen Enzalutamide and the impact on histone acetylation. Results are put into context doing an analysis of several public datasets.

Whereas the introduction itself is well structured and gives sufficient background to understand the research presented, there are some moderate language errors that should be corrected (e.g. "patient's progress" instead of "patients' progress" or "alternative spliced variants" instead of "alternatively spliced variants"). Overall, we recommend correction by a native speaker if not done already. It is also of note that the authors refer several times to "studies" and then present only a single reference (e.g. references 24 and 25 in lines 58 and 60).

The "Materials and Methods" part describes all methods used in the paper. In the description of the Seahorse metabolic flux assay, it is not clear which internal controls were used (e.g. to rule out any effect by albumin only or by the ethanol). The description of this method needs more detail. In the results, the Figure legend 2 can also not be understood with the information given in the methods section if you are not familiar with the assay. The section about the RT-PCR refers to the housekeeping gene as RPLA3 whereas it is called RPL13A in the supplemental methods. The nomenclature should be identical throughout the manuscript (also in the figure legends it is referred to as RPLA3). The description of Western Blot analysis is missing a reference ("...transferred to nitrocellulose (Invitrogen) as described."). In the supplemental methods, the primer for CPT1A is missing in the list of primers used in this paper.

In the "Results" part, the authors start with phenotypical analysis of CPT1A knockdown and overexpression in C42 cells. Of note, there is no description of those cells anywhere in the paper and it is not clear why this particular model was chosen. This becomes more important when the authors later use the cell line 22rv1 for comparison experiments and do not go into detail how these two cell lines are different (e.g. regarding the AR status). The effectiveness of knockdown and overxpression for the C42 cells is demonstrated on gene and protein expression level. Differences in clonogenic growth are compared. Figure 1E shows the results of a scratch wound assay to analyze migration. The result of this figure is not commented at all in the manuscript as the text only talks about the fact that it was not possible to analyze the OE cells in this assay. The authors should comment on the difference between KD cells and control cells at some point. Did the authors consider analyzing migration with a different type of assay (e.g. transwell migration assay) in order to also look at the OE cells? Furthermore, the authors analyzed growth in suspension with a sphere formation assay for the OE cells but not for the knockdown. Do these cells also show a tendency to form spheres? For the supplementary data, also brightfield pictures of all cells in 2D culture would be interesting. Do the cells show stem cell properties or signs of lineage differentiation?

Next, in Figure 2 mitochondria and lipid droplets are shown by immunofluorescence. Figure 2B and D show a quantitative analysis. In Figure 2B, there is a significant decrease of lipid droplets in the KD cells which is not mentioned anywhere, neither in the text nor the figure legends. The authors should comment on this. Figures 2E and F are the results of Seahorse assays. The abbreviation "OCR" is not explained anywhere. There are no controls shown for this experiment (e.g. only albumin). The authors should explain what controls were done for this experiment.

In Figure 3, the AR response is compared between the different stages of the C42 cells. It is shown that the AR response is increased in the KD cells and unchanged in the OE cells. Furthermore, on gene expression level an increase in total AR, AR full length and AR-V7 is visible in the knockdown cells. Here, protein expression analysis by Western Blot would be very interesting. To verify these results, knockdown and OE were also done in 22rv1. The results are shown in the supplementary material. Migration is analyzed only for controls and KD cells. How do the OE cells migrate? Do the authors see the same changes in adherence and spheroid formation seen for the C42 cells? Seahorse assay data is coherent with C42 cells. There is no data on differences in clonogenic growth of the 22rv1 cells, knockdown and OE are not validated by Western Blot and there is no data on lipid droplets and mitochondria. It would be interesting to see if the results from Figure 2A/C can be also reproduced in the second cell line. AR-fl and AR-V7 expression changes are completely different in 22rv1 as compared to C42. The authors should comment on this fact, perhaps taking into account the different AR status of these cell lines I mentioned earlier. Also, there is no data on AR target genes in 22rv1 anywhere in the manuscript.

In Figure 4, the effect of androgen stimulation and Enzalutamide treatment is analyzed. Strikingly, under DHT stimulation KD cells grow significantly better (both cell lines). C42 OE cells grow significantly worse under stimulation compared to controls. These cells also show resistance to Enzalutamide. There is no data on the Enzalutamide sensitivity of the KD cells. As these cells show the most significant change in AR expression (as shown in Figure 3C) it would be interesting to see how these cells behave under Enzalutamide treatment. Figures 4 D to F do not appear in the figure legend. This should be corrected.

In Figure 5 Enzalutamide treatment and tumor growth in castrated mice are then analyzed in xenograft experiments. KD cells show a decreased growth in the mice in general. OE cells however show a significantly higher growth under Enzalutamide treatment as compared to the controls. Here, it would be interesting to see the effect of Enzalutamide in uncastrated mice or at least cell culture experiments with Enzalutamide and CSS. Currently, the in vitro experiments are done with FBS (containing Testosterone) + Enzalutamide whereas the mouse experiments are done in a Testosterone-free environment which makes Figures 4 and 5 difficult to compare. The duration of treatment differs quite a lot between the groups shown in Figure 5 (between 14 and 32 days). Could the authors comment on that? Also, the scaling on the y axis (tumor volume) should be identical between the tumor growth picture (A/E) and the respective Enzalutamide treatment (e.g. 5A should have the same scaling as 5B, 5C same as 5D etc.).

In Figure 6 the authors analyze the effect of CPT1A KD and OE on lysine acetylation in FBS and CSS in C42 cells. Could the authors comment why they now chose this form of androgen deprivation which was not done in any of the previous experiments? It was shown that the overall level of histone acetylation increases under CSS conditions, especially in OE cells. The most dramatic effect is seen at H3K9ac, a mark for the active TSS. On the other hand, in FBS there is an increase of this mark in the KD and a decrease in the OE cells. The H3 loading control in FBS shown in Figure 6B is insufficient and should be re-done.

Next, mass spectrometry was done to further analyze the distribution of histone labeling. Here, it could be shown that labeled C13-acetate is used indeed in the de-novo synthesis of acetylated peptides which leads to the model proposed in Figure 6D. However, this does not explain the preference for H3K9ac and also does not explain the increased acetylation in FBS KD cells. Finally, cbioportal data is shown from several cohorts of PCA patient. Here, it could be shown that an increase of CPT1A is more prevalent in CRPC patients and associated with a worse prognosis.

In the "Discussion" the authors discuss the main findings from above. The authors propose a role for CPT1A inhibition in CRPC. They claim that enhanced H3K9ac in association with high CPT1A level is a predictor for Enzalutamide resistance. However, they do not comment on the changes of histone acetylation in the FBS group at all. To complete the last paragraph of the discussion, this should also be taken into consideration. Furthermore, the authors do not discuss the preference of one histone acetylation over others which is not coherent with the proposed model. The discussion should be extended in this direction.

Overall, I think the manuscript should be re-considered after the changes proposed here.

Author Response

Please see attachment, Thank you.

Reviewer 3 Report

The authors have conducted a logical series of studies that lead to their conclusion, namely that the lipid catabolism enzyme CPT1A plays a role in prostate cancer growth. The series of experiments detailed are clear and show he validity of the techniques employed to enhance or reduce CPT1A expression, and thus the suitability of utilization for in-vivo evaluations in murine models. 

However, there are a number of clarifications that are required to enhance the manuscript:

In the abstract, mention is made that prostate cancer that is metastatic is referred to as castrate resistant prostate cancer, it is actually referred to as being metastatic castrate resistant prostate cancer (mCRPC) In the introduction (line 29) ADT is references as being the standard treatment for localized and metastatic prostate cancer. Per NCCN guidleines, sipuelucel-T is considered first line treatment for mCRPC. Rationale for why the battery of experiments proving over/reduced expression of CPT1A was not performed on the 22Rv1 cell line On page 3 (line 126) mention is made of tumors growing to ~500CC.  Clarify if this is cubic centimeters - if so, this is not physiologically possible for a mouse. igure 2F/Page 5 refers to 'previous work demonstrating CPT1A KD cells having decreased ability to burn lipids.' The authors should have run the salient experiments and presented the results in this manuscript. Figure 4, panels C and F should be plotted in the same format as panels A, B, D and E

The authors rightfully mention that ADT is a treatment paradigm for prostate cancer - there is a body of data indicating that skewing towards a female hormone profile (i.e. testosterone reduction) may enhance immune system responsiveness and/or effect thymic functionality.  Thus, in an era of increased utilization of immunotherapies, and in light of the fact that an immunotherapy is approved for mCRPC the authors should provide some commentary on the role that CPT1A may play on the immune system.

Author Response

Please see attachment, Thank you.

Round 2

Reviewer 1 Report

The Ms has been extensively revised. Criticisms I raised have been clarified